# Study protocol for a phase 1, randomised, double-blind, placebo-controlled study to investigate the safety, tolerability and pharmacokinetics of ascending topical doses of TCP-25 applied to epidermal suction blister wounds in healthy male and female volunteers

Sigrid Lundgren [1,2] Karl Wallblom [1,2] Jane Fisher [1] Susanne Erdmann,[2] Artur Schmidtchen [1,2,3] Karim Saleh [1,2]

SL and KW contributed equally.

For numbered affiliations see end of article.

**Correspondence to**
Dr Sigrid Lundgren;
sigrid.lundgren@med.lu.se

## ABSTRACT

**Introduction** TCP-25 gel is intended for use in treatment of wound infection and inflammation. Current local therapies for wounds have limited efficacy to prevent infections and there are no wound treatments available today that target the excessive inflammation that often hampers wound healing in both acute and chronic wounds. There is therefore a high medical need for new therapeutic alternatives.

**Methods and analysis** A randomised, double-blinded, first-in-human study was designed to evaluate the safety, tolerability and potential systemic exposure of three increasing doses of the TCP-25 gel applied topically on suction blister wounds in healthy adults. The dose-escalation will be divided into three sequential dose groups with eight subjects in each group (24 patients in total). Within each dose group, the subjects will receive four wounds, with two wounds on each thigh. Each subject will receive TCP-25 on one wound per thigh and placebo on one wound per thigh in a randomised double-blinded manner, with a reverse reciprocal position on each respective thigh, to a total of five doses over 8 days. An internal safety review committee will monitor emerging safety and plasma concentration data over the course of the study and must give a favourable recommendation prior to initiating the next dose group, which will receive placebo gel or a higher concentration of TCP-25 in exactly the same manner described above.

**Ethics and dissemination** The study will be performed in accordance with ethical principles consistent with the Declaration of Helsinki, ICH/GCPE6 (R2), European Union Clinical Trials Directive and applicable local regulatory requirements.

This study is approved by the Swedish Medical Products Agency and the Swedish ethics committee under the registration number 2022-00527-01. The results of this study will be disseminated via publication to a peer-reviewed journal at the discretion of the Sponsor.

**Trial registration number** NCT05378997.

## STRENGTHS AND LIMITATIONS OF THIS STUDY

⇒ Wound induction through the suction blister technique is minimally invasive and provides a consistent, though limited, wound area.

⇒ Before each dose increase, an internal safety review committee evaluates the emerging safety data, ensuring utmost attention to safety.

⇒ Frequent outcome assessment is a scientific strength, but could limit the study as it is time-consuming and burdensome for participants.

⇒ This is a single-centre study with a limited number of participants, in a specific age range, thus limiting its applicability to other populations.

## INTRODUCTION

In normal conditions, wounds heal in a sequenced and timely manner, characterised by four major phases (haemostasis, inflammation, proliferation and remodelling). This is a complex process involving chemokines, growth factors, cytokines, proteases and antiproteases and multiple cell types that all work in a controlled manner during the healing process. This timely process can be interrupted at various stages and by different factors, leading to a dysfunctional inflammatory phase and disrupting wound healing. This common feature characterises most major wound complications.[1]

Wounds of various types have a significant impact on patients, healthcare and society.[1] Types of wounds include acute post-surgical wounds and burns, and non-healing wounds resulting from diabetes or circulatory disturbances.[2] It is critical both to prevent and

treat infections and to address excessive inflammation in various wound types. To our knowledge there is currently no available wound treatment that targets both the infection and the excessive inflammation simultaneously. Furthermore, as antimicrobial resistance increases, common antibiotics and some antiseptics are becoming even less effective in treating wound infections. Antimicrobial resistant bacteria are often present in various wounds, thus increasing the risk of antimicrobial resistant infections.[3–5] Therefore, there is a great need for new and cost-effective treatments that will improve healing and reduce both infection and inflammation in patients with various wounds.

TCP-25 is a new class of peptide-based drug which has a dual action targeting both infection and the associated excessive inflammation. A hydrogel containing TCP-25 treated infection and inflammation in murine and porcine wound infection models leading, hastening healing of infected wounds.[6 7] The mode of action is well characterised and involves interactions with bacterial lipopolysaccharide, other bacterial products and the cell-receptor CD14, enabling a downmodulation of inflammation, while simultaneously exerting direct antimicrobial effects. Importantly, TCP-25 acts on multiple multidrug resistant bacterial isolates.[6]

Suction blister wounds are standardised with respect to size, closure time and wound depth, as the epidermal layer always is ablated at the same level, exposing the dermis.[8] The mechanical non-bleeding excoriation elicits an innate immune response accompanied by increased blood flow and inflammatory cell infiltration in the underlying dermis. Epidermal regeneration occurs from the edges and appendages.[9] Usually, the wounds are colonised by commensal bacteria after some days.[10 11]

The safety and tolerability of three dose levels of TCP-25 gel will be tested to monitor safety in experimental wounds generated by the suction blister technique. It will be compared with a placebo control gel which is identical to TCP-25 gel, without the active ingredient (TCP-25). An overview of trial registration data is summarised in table 1.

## Study objectives

**The primary objective** of this study is to evaluate the safety and tolerability of three dose levels of TCP-25 gel (0.86 mg/mL, 2.9 mg/mL and 8.6 mg/mL) applied topically to epidermal suction blister wounds. **The secondary objective** is to evaluate the systemic exposure of TCP-25 after topical application to epidermal suction blister wounds.

**Exploratory objectives** include evaluation of wounds from photographs, and to collect and store samples from dressings and wound surfaces for future exploratory research. The exploratory analyses will be described in a separate report.

**Table 1** Trial registration data

| Data category | Information |
| --- | --- |
| Primary registry and trial identifying number | ClinicalTrials.gov NCT05378997 |
| Date of registration in primary registry | 18 May 2022 first submitted 25 April 2022. The trial was registered retrospectively by a few days because we experienced logistical delays in the registration of the all relevant parties in clinicaltrials.gov. |
| Secondary identifying numbers | EudraCT 2021-004728-14 (not public) |
| Source(s) of monetary or material support | Xinnate AB |
| Primary sponsor | Xinnate AB |
| Secondary sponsors | None |
| Contact for public queries | Karim Saleh, MD, PhD Email: karim.saleh@med.lu.se |
| Contact for scientific queries | Karim Saleh, MD, PhD (principal investigator) Email: karim.saleh@med.lu.se |
| Public title | Safety, Tolerability, and Pharmacokinetics of Ascending Topical Doses of TCP-25 Applied to Epidermal Suction Blister Wounds |
| Scientific title | A Phase 1, Randomized, Double-Blind, Placebo-Controlled Study in Healthy Male and Female Volunteers to Investigate the Safety, Tolerability, and Pharmacokinetics of Ascending Topical Doses of TCP-25 Applied to Epidermal Suction Blister Wounds |
| Countries of recruitment | Sweden |
| Health condition(s) or problem(s) studied | Acute epidermal wounds |
| Intervention(s) | Active comparator: TCP-25 gel (0.86, 2.9 or 8.6 mg/mL) |
| | Placebo comparator: placebo gel (equivalent gel with no TCP-25) |
| Key inclusion and exclusion criteria | Ages eligible for study: 18–60 years Sexes eligible for study: both Accepts healthy volunteers: yes |
| | Inclusion criteria: See table 2 |
| | Exclusion criteria: See table 2 |
| Study type | Interventional |
| | Allocation: Randomised Model: Sequential assignment (ascending doses) Masking: Quadruple (participant, care provider, investigator, outcomes assessor) |
| | Primary purpose: Treatment |
| | Phase I |

Continued

**Table 1** Continued

| Data category | Information |
|---|---|
| Date of first enrolment | 7 April 2022 |
| Target sample size | 24 |
| Recruitment status | Recruiting |
| Primary outcome(s) | Adverse events (time frame: 15 days)<br>Incidence of abnormal local reactions (local tolerability) (time frame: day 1, 2, 3, 5, 8, 11)<br>Number of patients with clinically significant changes from baseline in ECG, vital signs and physical examinations (time frame: During screening (baseline) and on day 11)<br>Number of patients with clinically significant changes from baseline in safety laboratory parameters (time frame: During screening (baseline) and on days 2, 3, 5, 11) |
| Key secondary outcomes | Plasma concentration of TCP-25 (time frame: day 1 (measured before blister formation), day 2 (measured before administration of the intervention and 0.5 and 1 hours after administration of the intervention), day 3 (measured before administration of the intervention 1 hour after administration of the intervention) and day 5 (measured before administration of intervention) |
| Ethics review | Approved 10 March 2022 Eti kprövningsmyndigheten registration number 2022-00527-01 |

## METHODS AND ANALYSIS

We used the Standard Protocol Items: Recommendations for Interventional Trials checklist when writing this report.[12]

### Trial design and setting

This trial is a randomised, double-blinded, first-in-human study designed to evaluate the safety, tolerability and potential systemic exposure of three increasing doses of the TCP-25 gel applied topically on suction blister wounds in healthy adults. The study will run as a single-centre study, conducted at Skåne University Hospital in Lund, Sweden. Screening and enrolment of the first dose group started on 7 April 2022. Randomisation and treatment administration started on 25 April 2022. The final follow-up for the last participant in the initial target sample size (24 participants) is estimated to be at the end of June 2022, with the possibility of extension beyond this date to include more participants. Data analysis will begin in April 2023. The dose-escalation will be divided into three sequential groups with eight subjects in each group. Within each dose group, each subject will receive TCP-25 on one wound on each thigh and placebo on one wound on each thigh in a randomised manner, but with a reverse reciprocal position on each respective thigh, to a total of five doses over 8 days. An internal safety review committee (iSRC) will monitor emerging safety and plasma concentration data over the course of the study and must give a favourable recommendation prior to initiating the higher dose in groups two and three, based on safety parameters as described below. Figure 1 presents an overview of the study design and which interventions and testing will be done on each day.

### Eligibility criteria

Prior to enrolment in the study, all subjects will be screened. Subjects will only be included if they fulfil all of

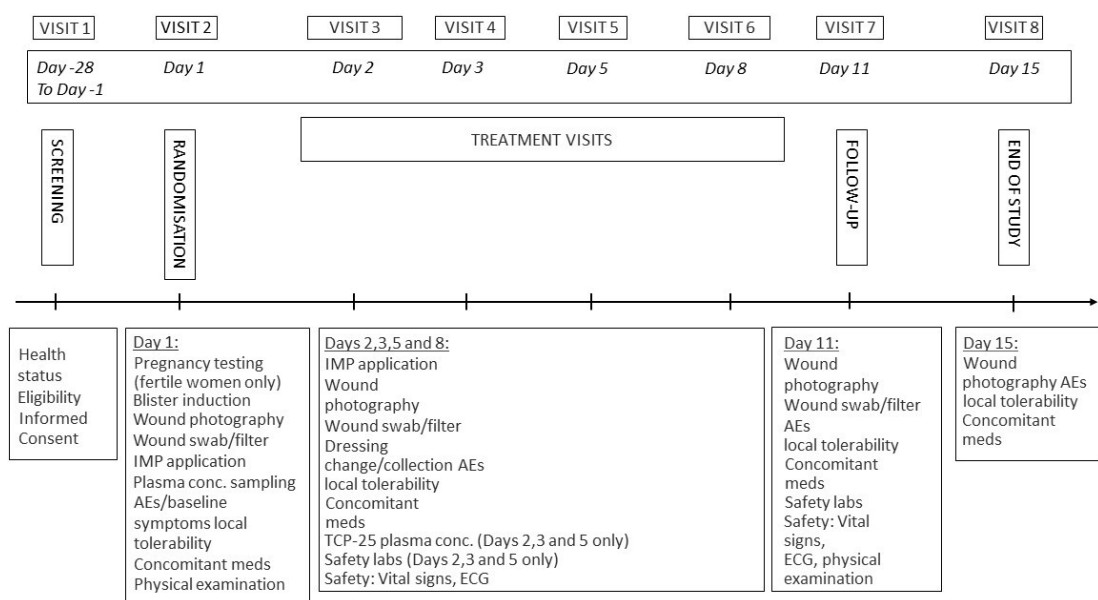

**Figure 1** Overview of the study design. Wound photography, wound swab/filter and dressing collection are samples to be collected and biobanked for future exploratory studies. AEs, adverse events; IMP, investigational medicinal product (TCP-25).

**Table 2** Summary of inclusion and exclusion criteria

| Inclusion criteria | Exclusion criteria |
|---|---|
| 1. Willing and able to give written informed consent for participation in the study.<br>2. Healthy male or female subject 18–60 years (inclusive) of age at the time of signing the informed consent.<br>3. Body mass index ≥18.0 and ≤30.0 kg/m$^2$.<br>4. Healthy and intact skin where the blister suction wounds will be induced.<br>5. Women of childbearing potential must have a documented negative serum pregnancy test done at the screening visit, within 4 weeks prior to suction blister formation and the start of study treatment. Male subjects must be willing to use condom or be vasectomised or practice sexual abstinence to prevent pregnancy.<br>6. Clinically relevant medical history, physical findings, vital signs, ECG and laboratory values at the time of screening. | 1. History of any clinically significant disease or disorder.<br>2. Any disease that may interfere with wound healing.<br>3. Active skin disease and/or tattoos in the areas where suction blister wounds will be induced.<br>4. Any planned major surgery within the duration of the study.<br>5. Any vital sign values outside the normal range.<br>6. Any clinically significant abnormalities in the resting ECG at the time of screening.<br>7. Current smokers or users of nicotine products.<br>8. Female subjects who are pregnant or lactating or planning a pregnancy.<br>9. Systemic immunosuppressive treatment.<br>10. Subjects who are currently receiving or have received antibiotics or systemic corticosteroids within 2 weeks prior to screening.<br>11. Regular use of anticoagulants or non-steroidal anti-inflammatory drugs within 2 weeks prior to the (first) administration of the intervention.<br>12. History of or ongoing severe allergy/hypersensitivity to drugs with a similar chemical structure or class to TCP-25 or to any excipients.<br>13. Planned treatment or treatment with another investigational drug within 3 months prior to day −1.<br>14. History of alcohol abuse or excessive intake of alcohol.<br>15. Presence or history of drug misuse.<br>16. Plasma donation within 1 month of screening or blood donation (or corresponding blood loss) during the 3 months prior to screening.<br>17. Involvement in the planning and/or conduct of the study.<br>18. Unlikely to comply with study procedures, restrictions and requirements. |

the inclusion criteria and none of the exclusion criteria. A summary of the eligibility criteria is listed in table 2. Details of the eligibility criteria, prior and concomitant therapy restrictions and general restrictions during the study can be found in online supplemental file 1–3.

## Consent and withdrawal

All subjects will be provided with written and oral information about the trial, including collection of biological specimens for exploratory studies, by the investigator or authorised associate and given sufficient time to make an informed decision about participation. The informed consent will be collected by a designated physician who is responsible for screening. Informed consent must be signed and documented before any study-specific procedures are performed.

Subjects will be free to discontinue their participation in the study at any time and for whatever reason without affecting their right to an appropriate follow-up investigation or their future care. If possible, the reason for withdrawal of consent shall be documented. Subjects may also be discontinued from the study at any time at the discretion of the Investigator according to the criteria described in online supplemental file 4. A subject who prematurely discontinues participation in the study will be asked about the presence of any adverse events and the Investigator must ask the subject if he/she is willing to be assessed according to the procedures scheduled for the end-of-study visit. Any ongoing adverse events will be followed until they are resolved.

## Suction blister wound formation

Blister wounds will be formed on the medial aspect of the thigh, two blisters on each leg with a 6 cm distance in between them. Before wounding, hair at the wound sites will be shaved. Each wound site will be wiped with an ethanol-soaked gauze. The exact wound sites will be marked using a skin marker.

The blisters will be made using the Model NP-4 (Electronic Diversities, Finksburg, Maryland, USA) suctioning device[13] operating in the negative pressure range 200–400 mm Hg with heating for optimal blister formation. Blister wound diameter will be 10 mm. The device will run until all blisters have formed, which takes approximately 60–70 min. The roof of the blister will be excised using sterile forceps and scissors.

The Model NP-4 suctioning device used in this study has the possibility to induce four wounds at the same time without causing notably more discomfort for the patient. We chose to make four wounds on each patient to increase the amount of the safety data we will obtain, and therefore the rigour of the study, without requiring the inclusion of more patients.

## Interventions

### Treatment

Each wound will receive 0.15 mL of a sterile gel containing either TCP-25 (0.86, 2.9 or 8.6 mg/mL according to the dose group) or placebo (0 mg/mL TCP-25), applied topically.

TCP-25 is a synthetic peptide based on the 25 C-terminal amino acids of human thrombin, with a molecular weight of 3088.6 Da.[6 7] TCP-25 is soluble in water and formulated in a proprietary hydrogel based on hydroxyethyl cellulose with glycerol included for isotonicity. pH is 7.0.

When the three chosen concentrations (0.86 mg/mL, 2.9 mg/mL and 8.6 mg/mL) are applied in a volume of

$0.15\,\text{mL}$ to a wound area of $0.8\,\text{cm}^2$, this results in total doses of $0.16\,\text{mg/cm}^2$, $0.54\,\text{mg/cm}^2$ and $1.6\,\text{mg/cm}^2$, respectively. These doses were based on effective doses identified in in vitro and in vivo tests.[6 7] Furthermore, the dose escalation steps were discussed with regulatory experts and selected to adequately assess safety and exploratory variables.

The preclinical safety of the drug product has been assessed through the necessary toxicity tests in mouse and porcine models, approved by the Swedish Medical Products Agency prior to initiation of this trial.

### Dressing

After application of the assigned intervention, each wound will be dressed using a $2\times2\,\text{cm}$ polyurethane foam (Mepilex Transfer, Mölnlycke Healthcare) as primary dressing and covered with a secondary dressing (Tegaderm, 3M). A secondary protective layer will be applied, consisting of overlapping gauze swabs covered with a secondary Tegaderm dressing.

### Outcomes

The follow-up period will be 15 days after application of the intervention. This short time period ensures participant retention and complete follow-up. The exact time points for evaluation of the following outcomes can be seen in figure 1. The methods for assessment are described under 'data collection' below.

The primary objective of safety and tolerability will be evaluated using the following:

### Primary outcomes

► Frequency, intensity and seriousness of adverse events (AEs) on days 1–15.
► Local tolerability by direct investigator evaluation of the incidence of abnormal local reactions as compared with expected wound healing outcome on days 1, 2, 3, 5, 8, 11 and 15, as described in the data collection section below.
► Clinically significant changes from baseline, as determined by the investigators, in the following measures (measured at baseline and on day 11):
  – ECG.
  – Vital signs (systolic/diastolic blood pressure, pulse rate).
  – Physical examinations (general condition, lymph nodes, throat, heart, lungs and abdomen).
► Clinically significant changes from baseline, as determined by the investigators, in safety laboratory parameters (haematology, clinical chemistry and coagulation, described in detail in table 3) at baseline and on days 2, 3, 5 and 11.

The secondary objective of systemic exposure will be evaluated using the following:

### Secondary outcome

► TCP-25 plasma concentrations as measured at day 1 (before blister formation), day 2 (before administration of the intervention and 0.5 and 1 hours after

administration of the intervention), day 3 (before administration of the intervention and 1 hour after administration of the intervention) and day 5 (before administration of the intervention).

### Data collection

All outcomes will be assessed by the investigator, who is a physician trained in Good Clinical Practice and qualified to assess these outcomes.

### Local tolerability

The investigator will evaluate tolerability by noting abnormal reactions by direct assessment of:
► Skin and wound erythema (abnormal reactions noted).
► Skin and wound oedema and swelling (abnormal reactions noted).
► Wound necrosis, crusting and haemorrhage (abnormal reactions noted).
► Wound purulent discharge as sign of excessive bacterial colonisation and/or infection.

The result (yes/no) for each parameter and time point will be documented in the electronic case report form

**Table 3** Safety laboratory parameters

| Category | Parameter |
| --- | --- |
| Clinical chemistry | Alanine aminotransferase |
| | Aspartate aminotransferase |
| | Creatinine |
| | C-reactive protein |
| | Glucose |
| | Glycated haemoglobin |
| Haematology | Haematocrit |
| | Haemoglobin |
| | Erythrocytes |
| | Mean corpuscular volume |
| | Mean corpuscular haemoglobin |
| | Mean corpuscular haemoglobin concentration |
| | Thrombocytes |
| | Leucocytes |
| | Eosinophils |
| | Neutrophils |
| | Basophils |
| | Lymphocytes |
| | Monocytes |
| Coagulation | Activated partial thromboplastin time Prothrombin complex/international normalised ratio |
| Pregnancy test (WOCBP only) | Serum pregnancy test at screening |
| | Urine pregnancy test (visit 2) |

WOCBP, women of childbearing potential.

(eCRF). Local tolerability parameters, which are judged by the investigator to be aggravated compared with what is expected for blister-induced epidermal wounds, will be reported as AEs.

## ECG

Single 12-lead ECG will be recorded in supine position after 10 min of rest using an ECG machine. Heart rate, PR, QRS and QT intervals and corrected QT interval by Fredericia will be recorded. Safety ECGs will be reviewed and interpreted on-site by the investigator. Any abnormalities will be specified and documented as not clinically significant or clinically significant. Abnormal post-dose findings assessed by the investigator as clinically significant will be reported as AEs.

## Vital signs

Will be assessed in the sitting position after 10 min of rest. Any vital signs outside the normal ranges will be judged as not clinically significant or clinically significant. The assessment will be recorded in the eCRF. Vital signs judged by the investigator as 'abnormal, clinically significant' after application of the intervention will be reported as AEs.

## Physical examination

A physical examination will include assessments of general condition, lymph nodes, throat, heart, lungs and abdomen. Any abnormalities will be specified and documented as clinically significant or not clinically significant. Abnormal post-dose findings assessed by the Investigator as clinically significant will be reported as AEs.

## Safety laboratory measurements

Blood will be collected with a peripheral venous catheter using the standard clinical operating procedure and measured by the clinical chemistry department at Skåne University Hospital, Lund, using standard clinical assays. The safety laboratory parameters measured are detailed in table 1.

## AEs

The grading of the intensity of AEs will follow the Common Terminology Criteria for Adverse Events V.5.0.[14] All AEs, including those AEs defined above, shall be recorded. AE will be assessed for causality and severity by the investigator, and any suspected unexpected severe adverse reactions will be reported to the local competent authority and ethics committee according to local and European regulations.

## TCP-25 plasma concentrations

Samples for determination of plasma concentrations of TCP-25, will be analysed by Q&Q Laboratories AB, Göteborg, Sweden, by means of a validated liquid chromatography - tandem mass spectrometry method. The lower limit of quantification is 30 nmol/L.

These interventions and measurements will be done according to the schedule presented in figure 1.

## Sample size and recruitment

No formal sample size calculation has been performed for this study because the study objectives involve safety with few events expected. The proposed sample size is considered sufficient to provide adequate information for the study objectives. Approximately 50 subjects will be screened to achieve 24 randomised and evaluable subjects. The subjects will be healthy volunteers recruited from the Clinical Trial Unit's database of healthy volunteers, using the Unit's regular routines for recruiting. Briefly, potential candidates are contacted by email or phone and provided information about the study. Candidates who sign up for the study are then contacted with further information, followed by screening as described above. The text approved by the independent ethics committee (online supplemental file 5, in Swedish) will be used to create all the materials for recruitment.

## Allocation

Each subject will have four suction blister wounds, two on each thigh. Wounds will be labelled R1 (right thigh, distal), R2 (right thigh, proximal), L1 (left thigh, distal) and L2 (left thigh, proximal).

A computer-generated randomisation list will be created using SAS Proc Plan, SAS V.9.4 by a designated responsible person at Clinical Trial Consultants AB. The randomisation list will contain subject number, wound position, thigh (left/right) and treatment and will be kept by the randomiser in a sealed envelope until database lock. On day 1 the subjects in each dose group (n=8) will be randomised with regards to the treatment allocation of each wound. Two wounds per patient will receive TCP-25, and the other two will receive placebo. The proximal wound (R1 or L1) of one thigh and the distal wound of the other thigh (L2 or R2) will receive the same treatment, according to the assigned treatment allocation.

An unblinded research nurse in a nearby medical room will read the randomisation list, which indicates the assigned allocation of TCP-25 and control gel to the various wounds for each patient. The research nurse will then fill two syringes per patient. One syringe will contain TCP-25 gel and the other will contain placebo gel. Both syringes will be labelled with the randomisation number of the patient. Additionally they will be labelled with either 'R1+L2' or 'R2+L1', corresponding to the two wounds (right proximal+left distal or right distal+left proximal) that are to receive the contents of that syringe according to the assigned treatment allocation. These prefilled and labelled syringes will be provided to the blinded research staff who will then deposit their contents onto the correct combination of wounds indicated on the syringe label. In this way, the allocation will be communicated to the research staff while ensuring that they remain blinded to the treatment given to each wound.

If needed for emergency unblinding, sealed individual treatment code envelopes will be kept at the clinic and at a contract research organisation (Clinical Trial Consultants

AB's Pharmacovigilance department) in locked and restricted areas.

## Blinding

This is a double-blind study and the allocation of treatments will not be disclosed until a clean file has been declared and the database has been locked. The TCP-25 and placebo gels are identical in appearance.

**Unblinding** will only happen in the following situations:
1. The treatment code may only be broken by the Principal Investigator or delegate in case of emergency when knowledge of the treatment received is necessary for the proper medical management of the subject.
2. The randomisation code may be broken by the iSRC during the assessment process (partial unblinding) to enable their decision on continued dosing of further cohorts. The medical staff and subjects will remain blinded to the treatment allocation.

## Data management

The data management routines include procedures for handling of the eCRF, database set-up and management, data entry and verification, data validation, quality control of the database and documentation of the performed activities including information of discrepancies in the process. Data validation/data cleaning procedures are designed to assure validity and accuracy of clinical data and consist of computerised online edit checks such as range checks and batch checks on data exports. All study-specific and standard data validation programming will be tested in a separate testing environment prior to use on production data. Detailed information on data management is described in a study-specific Data Management Plan.

Clinical data will be entered into an eCRF provided by Viedoc Technologies AB. Clinical data will be entered directly from the source documents or at bedside by authorised site personnel designated by the Investigator. All entries of data, corrections and alterations in the eCRF are to be made by the Investigator or designee, who is to be a qualified physician trained in Good Clinical Practice.

Because data entry is single, all data will undergo source data verification by monitors according to a risk-based monitoring plan. Monitors will review the eCRFs for completeness and consistency and compare data with the respective source documents. If corrections are needed, queries will be raised within the eCRF, either as a result of built-in edit checks or manually and corrected by the site staff. When all data have been entered and discrepancies solved, a clean file will be declared, the database will be locked and the code will be broken. The data analysis will begin in April 2023.

## Statistical analysis plan

Continuous data will be presented in terms of evaluable and missing observations, arithmetic mean, SD, median, minimum and maximum value. Categorical data will be presented as counts and percentages. When applicable, summary data will be presented by treatment, and by assessment time. Individual subject data will be listed by subject number, treatment and, where applicable, by assessment time. All descriptive summaries and statistical analyses will be performed using SAS V.9.4 or later (SAS Institute). Baseline will be defined as the last non-missing observation prior to the first administration of the intervention.

Outliers will be included in summary tables and listings and will not be handled separately in any analyses. Generally, no imputation of missing data will be performed. However, clinical safety laboratory parameters that are outside the detection limit, will be replaced with the detection limit when calculating statistics. Also, when calculating statistics for pharmacokinetics plasma concentrations, concentrations under lower limit of quantification will be replaced with lower limit of quantification if more than 50% of the values for a given time point is above lower limit of quantification. Otherwise, no statistics will be calculated for that time point.

## Data monitoring
### iSRC

Before initiating a new dose group, all subjects in the previous dose group must have been treated and safety, tolerability and pharmacokinetics data (at least data up to 5 days for pharmacokinetics and 8 days for safety) for all treated subjects must have been evaluated by the iSRC. In case of non-replaced dropouts, available data for the dropouts will be included in the iSRC evaluation.

If TCP-25 is considered safe and tolerable, a written recommendation on the next dose level will be provided to the Sponsor.

The voting members of the iSRC will consist of the principal investigator or delegate and the chief physician at the Clinical Trial Unit or delegate. In addition, the study clinical research manager, the study pharmacokineticist and additional Sponsor representatives may be invited as appropriate. Further internal or external experts may be invited and consulted by the iSRC as appropriate.

The recommendation to the Sponsor may be study progression, to continue with a higher or lower dose than the intended dose, to repeat the same dose level, to continue with an intermediate dose level or to stop dosing. The recommendation of the iSRC on the next dose level will be made in consensus between the iSRC members and documented as appropriate.

In case there is disagreement between the two voting members, the most conservative approach will be taken.

## Auditing

The Sponsor has employed a contract research organisation (Clinical Trial Consultants AB) as an independent monitor. Monitoring will be done according to a risk-based monitoring plan and the responsible monitor will periodically visit the study site at times agreed on by the investigator and the monitor. At each monitoring visit, the role of the Monitor includes, but is not limited to:

- ► Confirm that facilities and resources remain acceptable.
- ► Confirm that the investigational team is adhering to the study protocol, applicable standard operating procedures, guidelines, manuals and regulatory requirements.
- ► Verify that data are being recorded in the eCRFs in an accurate and timely manner and that accountability checks are being performed.
- ► Verify that data in the eCRF are consistent with the clinical records in accordance with the monitoring plan.
- ► Verify that the correct informed consent procedure has been adhered to for participating subjects.
- ► Verify that AEs are recorded and reported in a timely manner and according to the clinical study protocol.
- ► Raise and escalate any serious quality issues, serious Good Clinical Practice breach and any data privacy breach to the Sponsor.

Centralised monitoring will also be performed continuously by study team members at Clinical Trial Consultants AB in accordance with the monitoring plan. When the study has been completed and all queries have been resolved and the database has been locked, the Monitor will perform a close-out visit.

Authorised representatives of the Sponsor, a competent authority or an independent ethics committee may also perform audits or inspections at the research clinic, including source data verification.

## ETHICS AND DISSEMINATION
### Ethical conduct of the study
The study will be performed in accordance with ethical principles that have their origin in the Declaration of Helsinki and are consistent with ICH/GCPE6 (R2), European Union (EU) Clinical Trials Directive and applicable local regulatory requirements. The principal investigator is responsible for submitting the study protocol, the subject information and informed consent form, any other written information to be provided to the subjects and any advertisements used for subject recruitment to the Swedish Ethical Review Authority for approval. The study has been approved by the Swedish Ethical Review Authority (registration number 2022-00527-01).

Before performing any study-related procedures the informed consent form shall be signed and personally dated by the subject and by the investigator. The subjects will be covered under the Sponsors liability insurance policy. The participating subjects are also protected in accordance with national regulations, as applicable.

### Subject data protection
The informed consent form includes information, that data will be recorded, collected and processed and may be transferred to other European Economic Area (EEA) or non-EEA countries. In accordance with the General Data Protection Regulation (EU) 2016/679, the data will not identify any persons taking part in the study.

### Clinical study report
A summarising report shall be submitted to the Swedish competent authority and independent ethics committee within 12 months after completion of the study. A clinical study report, in compliance with ICH-E3, describing the conduct of the study, any statistical analyses performed and the results obtained, will be prepared. The report will be reviewed and approved by, as a minimum, the principal investigator, the statistician and the Sponsor. The study results will be reported in the EudraCT database per applicable regulations within 12 months after completion of the study. All data obtained from any exploratory analyses will be reported separately.

### Publication and dissemination
Publication of the study results and access to the study data is at the discretion of the Sponsor. Authorship for any publications will be according to the Vancouver criteria. Subjects will be asked to provide a release for use of their photographs in publications if applicable. The subject has a right to refuse the photo release without jeopardising their eligibility to participate in the study.

### Patient and public involvement
Patients or the public were neither involved in the design nor will they be involved in this clinical trial's conduct, reporting or dissemination plans.

### Protocol version
Protocol V.2.0; 25 January 2022. Protocol amendments will be promptly communicated to all relevant parties, trial registries and to the ethics committee and competent authority as required by law.

**Author affiliations**
[1]Section for Dermatology and Venereology, Department of Clinical Sciences Lund, Lund University, Lund, Sweden
[2]Department of Dermatology, Skåne University Hospital Lund, Lund, Sweden
[3]Copenhagen Wound Healing Center, Bispebjerg Hospital, Department of Biomedical Sciences, University of Copenhagen, Copenhagen, Denmark

**Acknowledgements** We acknowledge the support of the staff at the Clinical Trial Unit at Skane University Hospital Lund and Clinical Trial Consultants AB for input on the Clinical Study Protocol. Xinnate AB provided the project management resources and additional expertise for the regulatory development of TCP-25 gel and the completion of the Clinical Study Protocol. We thank Anne Nielsen and other personnel at the Department of Dermatology Lund and the Clinical Trial Unit at Skane University Hospital Lund for support with the clinical parts of the study. We used the Standard Protocol Items: Recommendations for Interventional Trials checklist when writing this report.

**Contributors** SL prepared the first draft of the manuscript, contributed to the development of the clinical study protocol and revised and approved the final manuscript. KW prepared the first draft of the manuscript, contributed to the development of the clinical study protocol and revised and approved the final manuscript. JF contributed to the development of the clinical study protocol and revised and approved the final manuscript. SE contributed to the development of the clinical study protocol and revised and approved the final manuscript. AS was responsible for the conception of the study, contributed to the development of the clinical study protocol and revised and approved the final manuscript. KS was

responsible for the conception of the study, contributed to the development of the clinical study protocol and revised and approved the final manuscript.

**Funding** The Safety study described here was funded by Xinnate AB. Exploratory analyses on data and materials originating from this study are integrated in other research projects funded by research grants from the Swedish Research Council (project 2020-02016), Edvard Welanders Stiftelse and Finsenstiftelsen (Hudfonden), the Österlund Foundation, and the Swedish Government Funds for Clinical Research (ALF).

**Competing interests** AS is a co-founder and shareholder of Xinnate AB. The other authors report no competing interests.

**Patient and public involvement** Patients and/or the public were not involved in the design, or conduct, or reporting, or dissemination plans of this research.

**Patient consent for publication** Not applicable.

**Provenance and peer review** Not commissioned; externally peer reviewed.

**ORCID iDs**
Sigrid Lundgren http://orcid.org/0000-0001-6693-0233
Karl Wallblom http://orcid.org/0000-0003-0947-1450
Jane Fisher http://orcid.org/0000-0002-3780-901X
Artur Schmidtchen http://orcid.org/0000-0001-9209-3141
Karim Saleh http://orcid.org/0000-0003-0604-5739

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
