## [Reviewer comments · BMJ Open]

ARTICLE DETAILS

TITLE (PROVISIONAL)	Study protocol for a Phase 1, Randomized, Double-Blind, Placebo-Controlled Study to Investigate the Safety, Tolerability, and Pharmacokinetics of Ascending Topical Doses of TCP-25 Applied to Epidermal Suction Blister Wounds in Healthy Male and Female Volunteers
AUTHORS	Lundgren, Sigrid; Wallblom, Karl; Fisher, Jane; Erdmann, Susanne; Schmidtchen, Artur; Saleh, Karim

VERSION 1 – REVIEW

REVIEWER	Arundel, Catherine University of York
REVIEW RETURNED	15-Aug-2022

GENERAL COMMENTS	This is a well written paper which clearly follows SPIRIT guidance. A few points of clarification are suggested below: - It would be helpful to detail who takes informed consent and also to add the timing of informed consent to Figure 1-It would be helpful to add additional detail about how the study outcomes are assessed, by who and when.- It was also unclear what the primary outcome was for the study, safety or tolerability, and i suggest it would be worth rephrasing to study outcomes rather than study endpoints.- Some additional detail on the types or recruitment advertisements and where these would be used would be helpful- Please confirm the type of randomisation used in this study. Please also detail who generated the randomisation sequence. It was also not completely clear how allocations would be notified to relevant individuals so some amendment here would be helpful.- Have any strategies to support retention been considered?- Who completes the eCRFs for the study?- Is data entry single or double? If single, is a proportion checked at all?
--

REVIEWER	Ågren, Magnus Københavns Universitet, Clinical Medicine
REVIEW RETURNED	15-Oct-2022

GENERAL COMMENTS	1. Page 2, Abstract, lines 12-17: Design, allocation of treatments and definition of groups (e.g., groups 2 and 3 are not described) need clarification.
--

	2. Abstract, line 25: I was unable to retrieve the trial with given number provided (EudraCT 2021-004728-14) on https://www.clinicaltrialsregister.eu 3. Page 4, line 4: "... exposed, size and ..." Please check this sentence for grammar. 4. Page 4, line 8: Please cite the paper of Burian 2022 (PMID: 35578822) also. 5. Page 4, lines 30-32: I am confused here; did the investigators already include participants? If so, please provide data on them. Were (or will be) equal numbers of males and females included? 6. Page 6, line 4: Please convert to the SI unit (mm Hg). 7. Page 6, lines 9-11: Please provide more data on the compound TCP-25 and formulation such as molecular size, pH, physicochemical properties and composition of the hydrogel vehicle. Please, give the scientific reason for testing the rather odd concentration and the dose intervals of TCP-25. Are there safety issues connected with the hydrogel or has it been sufficiently assessed? Also, indicate whether there are differences in terms of appearance, smell, etc that can jeopardize the blinding. 8. Page 6, line 13: What material is Mepilex transfer made of? 9. Page 6, line 16: Can the drug induce pain at application (see Larsen et al. (11) for details on assessments). If so, there is a potential risk of ruining the blinding. 10. Page 8, line 20: What is the rational for making and treating two wounds on each thigh?
--	--

REVIEWER	Sjoberg, Folke Linkopings universitet
REVIEW RETURNED	25-Oct-2022

GENERAL COMMENTS	MS: BMJ Open; Article type: Protocol Study protocol for a Phase 1, Randomized, Double-Blind, Placebo-Controlled Study to Investigate the Safety, Tolerability, and Pharmacokinetics of Ascending Topical Doses of TCP-25 Applied to Epidermal Suction Blister Wounds in Healthy Male and Female Volunteers. This is a protocol manuscript describing a First in human Phase 1 randomized trial to access safety and tolerability of ascending doses of TCP-25 25 applied to epidermal suction blister wounds in healthy male and female volunteers. This is a well-planned, clearly written protocol of the study. The manuscript reads well and has attention to detail. There are a few topics that if added to the manuscript may improve it. First, the manuscript would merit adding the dose levels selected. The rationale for the concentration, the volume and dose/cm² Second, adding a comment on the number of blisters per healthy volunteer, and the number of volunteers selected? More wounds on fewer healthy volunteers? Third, is there really a reason to believe a systemic absorption is relevant? Detection resolution of the technique for the assessment of the blood concentration? Please expand and comment. Fourth, a screen failure rate of 50% is anticipated. It appears high? Any reasons for this high rate?
---

VERSION 1 – AUTHOR RESPONSE

Reviewer: 1 Dr. Catherine Arundel, University of York Comments to the Author:
This is a well written paper which clearly follows SPIRIT guidance.

A few points of clarification are suggested below:

- It would be helpful to detail who takes informed consent and also to add the timing of informed consent to Figure 1

We have now added that "The informed consent will be collected by a designated physician who is responsible for screening." (page 5). We have also added the timing of the informed consent to Figure 1 as suggested.

-It would be helpful to add additional detail about how the study outcomes are assessed, by who and when.

We have now clarified the timepoints for assessment of each outcome in the "outcomes" section (page 7) and referred to figure 1 for a visual overview of the timing of outcome assessment relative to the other study procedures.

Details about how the outcomes will be assessed and by whom are in the "data collection" section (page 8). We have also clarified that "All outcomes will be assessed by the investigator, who is a physician trained in good clinical practice and qualified to assess these outcomes." (page 8)

- It was also unclear what the primary outcome was for the study, safety or tolerability, and I suggest it would be worth rephrasing to study outcomes rather than study endpoints.

We have now rephrased them as outcomes instead of endpoints. Because this was a phase I study, we had several different primary outcomes which are considered equally important and are all listed in the "primary outcomes" section (page 7).

- Some additional detail on the types or recruitment advertisements and where these would be used would be helpful

We have now clarified that "The subjects will be healthy volunteers recruited from the Clinical Trial Unit's database of healthy volunteers, using the Unit's regular routines for recruiting. Briefly, potential candidates are contacted by email or phone and provided information about the study. Candidates who sign up for the study are then contacted with further information, followed by screening as described above. The text approved by the independent ethics committee (supplement 5, in Swedish) will be used to create all the materials for recruitment." (page 9)

- Please confirm the type of randomisation used in this study. Please also detail who generated the randomisation sequence. It was also not completely clear how allocations would be notified to relevant individuals so some amendment here would be helpful.

We have now clarified the wording in the "allocation" section (page 10) to make it more clear who generates the randomisation sequence and how the allocations will be notified to the research staff. The new wording is as follows:

"A computer-generated randomisation list will be created using SAS Proc Plan, SAS Version 9.4 by a designated responsible person at Clinical Trial Consultants AB."

"An unblinded research nurse in a nearby medical room will read the randomisation list, which indicates the assigned allocation of TCP-25 and control gel to the various wounds for each patient. The research nurse will then fill two syringes per patient. One syringe will contain TCP-25 gel and the other will contain placebo gel. Both syringes will be labelled with the randomisation number of the patient. Additionally they will be labelled with either "R1+L2" or "R2+L1", corresponding to the two wounds (right proximal + left distal or right distal + left proximal) that are to receive the contents of that syringe according to the assigned treatment allocation. These pre-filled and labelled syringes will be provided to the blinded research staff who will then deposit their contents onto the correct combination of wounds indicated on the syringe label. In this way, the allocation will be communicated to the research staff while ensuring that they remain blinded to the treatment given to each wound."

- Have any strategies to support retention been considered?

Due to the short follow-up period, we do not anticipate any difficulties with retention and anticipate little to no loss to follow-up. Therefore we have not considered any strategies to support retention.

- Who completes the eCRFs for the study?

As we wrote in the "data management" section, "all entries of data, corrections, and alterations in the

eCRF are to be made by the Investigator or designee". For privacy reasons, and to allow flexibility in appointing additional designees as needed, we do not wish to name the person who is currently appointed as the investigator or designee in this publication, however we have clarified that this individual "is to be a qualified physician trained in good clinical practice" (page 11).

- Is data entry single or double? If single, is a proportion checked at all?

We have now clarified that "Because data entry is single, data will undergo source data verification by monitors according to a risk-based monitoring plan." (page 11)

Reviewer: 2 Dr. Magnus Ågren, Københavns Universitet Comments to the Author:

1. Page 2, Abstract, lines 12-17: Design, allocation of treatments and definition of groups (e.g., groups 2 and 3 are not described) need clarification.

We have now clarified the wording of the design, allocation, and group definitions to read "The dose-escalation will be divided into 3 sequential dose groups with 8 subjects in each group (24 patients in total). Within each dose group, the subjects will receive four wounds, with two wounds on each thigh. Each subject will receive TCP-25 on one wound on each thigh and placebo on one wound on each thigh in a randomised double-blinded manner, but with a reverse reciprocal position on each respective thigh, to a total of 5 doses over 8 days. An internal safety review committee will monitor emerging safety and plasma concentration data over the course of the study and must give a favourable recommendation prior to initiating the next dose group, which will receive placebo gel or a higher concentration of TCP-25 in exactly the same manner as described above."

2. Abstract, line 25: I was unable to retrieve the trial with given number provided (EudraCT 2021-004728-14) on <https://www.clinicaltrialsregister.eu>

Unfortunately Phase 1 trials registered in EudraCT are not publicly available in Europe. Therefore, to ensure public registration, we also registered it on clinicaltrials.gov (NCT05378997). We have now clarified in table 1 that the EudraCT registration is not public.

3. Page 4, line 4: "... exposed, size and ..." Please check this sentence for grammar.

Thank you for catching this, we have now reworded it to read "Suction blister wounds are standardised with respect to size, closure time, and wound depth, as the epidermal layer always is ablated at the same level, exposing the dermis (8)."

4. Page 4, line 8: Please cite the paper of Burian 2022 (PMID: 35578822) also.

Thank you, we have now added this excellent citation.

5. Page 4, lines 30-32: I am confused here; did the investigators already include participants? If so, please provide data on them. Were (or will be) equal numbers of males and females included? When this manuscript was submitted for publication, screening and inclusion for the first dose group was completed; however no subjects were randomized or treated yet at that time. As this study was planned to run over several months, the screening and inclusion of participants for the consecutive dose groups were not initiated at the time of submission as screening was not allowed more than 30 days in advance of inclusion. Due to delays at BMJ Open following our submission, the inclusion of all subjects is actually complete at this time. However since this is a study protocol, it would not be appropriate to include any data from the study as this would result in double-publication of the data when we publish the study results later on. Therefore we will not include any data in this publication. There was no formal requirement in the study to include equal numbers of females and males in each group; however, we tried our best to achieve an equal distribution during the screening.

6. Page 6, line 4: Please convert to the SI unit (mm Hg).

We have now converted the units to 200-400 mmHg (page 6).

7. Page 6, lines 9-11: Please provide more data on the compound TCP-25 and formulation such as molecular size, pH, physicochemical properties and composition of the hydrogel vehicle.

We have added details about the TCP-25 compound and referred to studies where it is described in more detail. The formulation of the hydrogel is proprietary, however we have added some details to the text to the extent that we are able: "TCP-25 is a synthetic peptide based on the 25 C-terminal amino acids of human thrombin, with a molecular weight of 3088.6 Da (6, 7). TCP-25 is soluble in

water and formulated in a proprietary hydrogel based on hydroxyethyl cellulose with glycerol included for isotonicity. pH is 7.0.” (page 6)

Please, give the scientific reason for testing the rather odd concentration and the dose intervals of TCP-25.

We have added a rationale for the chosen dose intervals as follows “When the three chosen concentrations (0.86 mg/mL, 2.9 mg/mL and 8.6 mg/mL) are applied in a volume of 0.15ml to a wound area of 0.8 cm², this results in total doses of 0.16mg/cm², 0.54mg/cm² and 1.6mg/ cm² respectively. These doses were based on effective doses identified in in vitro and in vivo tests (6, 7). Furthermore, the dose escalation steps were discussed with regulatory experts and selected to adequately assess safety and exploratory variables.” (page 6)

Are there safety issues connected with the hydrogel or has it been sufficiently assessed?

We have now specified that “The pre-clinical safety of the drug product has been assessed through the necessary toxicity tests in mouse and porcine models, approved by the Swedish Medical Products Agency prior to initiation of this trial.” (page 6)

Also, indicate whether there are differences in terms of appearance, smell, etc that can jeopardize the blinding.

As we wrote in the “Blinding” section (page 10), “The TCP-25 and placebo gels are identical in appearance.”

8. Page 6, line 13: What material is Mepilex transfer made of?

We have now specified that it is a polyurethane foam.

9. Page 6, line 16: Can the drug induce pain at application (see Larsen et al. (11) for details on assessments). If so, there is a potential risk of ruining the blinding.

TCP-25 is an endogenous molecule that is naturally present in wounds. It also has a large molecular weight, which reduces its diffusion through the wound bed. Therefore, the risk of pain induction in the TCP-25 treated wounds is extremely minimal and should not risk unblinding. In future studies we will consider including pain as an outcome to be able to quantify this risk.

10. Page 8, line 20: What is the rationale for making and treating two wounds on each thigh?

We have now added a rationale for this as follows “The Model NP-4 suctioning device used in this study has the possibility to induce 4 wounds at the same time without causing notably more discomfort for the patient. We chose to make 4 wounds on each patient to increase amount of the safety data we will obtain, and therefore the rigour of the study, without requiring the inclusion of more patients.” (page 6)

Reviewer: 3 Dr. Folke Sjöberg, Linköpings universitet Comments to the Author:

This is a well-planned, clearly written protocol of the study. The manuscript reads well and has attention to detail. There are a few topics that if added to the manuscript may improve it.

First, the manuscript would merit adding the dose levels selected. The rationale for the concentration, the volume and dose/cm²

We have now added the dose levels and rationale as follows “When the three chosen concentrations (0.86 mg/mL, 2.9 mg/mL and 8.6 mg/mL) are applied in a volume of 0.15ml to a wound area of 0.8 cm², this results in total doses of 0.16mg/cm², 0.54mg/cm² and 1.6mg/ cm² respectively. These doses were based on effective doses identified in in vitro and in vivo tests (6, 7). Furthermore, the dose escalation steps were discussed with regulatory experts and selected to adequately assess safety and exploratory variables.” (page 6)

Second, adding a comment on the number of blisters per healthy volunteer, and the number of volunteers selected? More wounds on fewer healthy volunteers?

We have now added a rationale for this as follows “The Model NP-4 suctioning device used in this study has the possibility to induce 4 wounds at the same time without causing notably more discomfort for the patient. We chose to make 4 wounds on each patient to increase amount of the safety data we will obtain, and therefore the rigour of the study, without requiring the inclusion of more patients.” (page 6)

Third, is there really a reason to believe a systemic absorption is relevant?

TCP-25 is not expected to be systemically absorbed due to its molecular weight of 3 kDa. Plasma TCP-25 levels were monitored in a minipig toxicity study following application of TCP-25 (0.86 and 8.6 mg/mL for 28 days [1 mL/wound to 4 wounds at 9 cm² per animal]) to surgically established wounds. No plasma samples had concentrations above the LLOQ (30 nmol/L) showing that there was no systemic exposure during the study. Moreover, no systemic cardiovascular effects were observed in the porcine toxicity study.

However, as this is a first-in-human study of TCP-25 we are required to test for systemic absorption of the compound.

Detection resolution of the technique for the assessment of the blood concentration? Please expand and comment.

We have now specified that "Samples for determination of plasma concentrations of TCP-25, will be analysed by Q&Q Laboratories AB, Göteborg, Sweden, by means of a validated LC-MS/MS method. The LLOQ is 30 nmol/L." (page 9)

Fourth, a screen failure rate of 50% is anticipated. It appears high? Any reasons for this high rate?

The main reason why we estimate such a high rate is to be able to screen an excess number of volunteers in order to ensure that we will have a sufficient number of included participants, plus some backup participants in case of dropouts.

VERSION 2 – REVIEW

REVIEWER	Arundel, Catherine University of York
REVIEW RETURNED	13-Dec-2022
GENERAL COMMENTS	Thank you for attending to my suggestions for revision - I can confirm these have been made accordingly.
REVIEWER	Sjoberg, Folke Linkopings universitet
REVIEW RETURNED	14-Dec-2022
GENERAL COMMENTS	The Authors have adequately addressed all concerns of the reviewers.